# Effect of prorenin peptide vaccine on the early phase of diabetic retinopathy in a murine model of type 2 diabetes

**Harumasa Yokota**[1]◯*, **Hiroki Hayashi**[2]◯, **Junya Hanaguri**[1], **Satoru Yamagami**[1], **Akifumi Kushiyama**[3], **Hironori Nakagami**[2], **Taiji Nagaoka**[1]

**1** Division of Ophthalmology, Department of Visual Science, Nihon University School of Medicine, Tokyo, Japan, **2** Department of Health Development and Medicine, Osaka University, Osaka, Japan, **3** Department of Pharmacotherapy, Meiji Pharmaceutical University, Tokyo, Japan

◯ These authors contributed equally to this work.
\* atokoy18@gmail.com

**Data Availability Statement:** All relevant data are within the manuscript.

## Abstract

Prorenin is viewed as an ideal target molecule in the prevention of diabetic retinopathy. However, no drugs are available for inhibiting activation of prorenin. Here, we tested the effect of a prorenin peptide vaccine ($V_P$) in the retina of a murine model of type 2 diabetes (T2D). To choose the optimal vaccine, we selected three different epitopes of the prorenin prosegment (E1, E2, and E3) and conjugated them to keyhole limpet hemocyanin (KLH). We injected C57BL/6J mice twice with KLH only (as a control vaccine), E1 conjugated with KLH (E1-KLH), E2-KLH, or E3-KLH and compared antibody titers. E2-KLH showed the highest antibody titer and specific immunoreactivity of anti-sera against prorenin, so we used E2-KLH as $V_P$. Then, we administered injections to the non-diabetic db/m and diabetic db/db mice, as follows: db/m + KLH, db/db + KLH, and db/db + $V_P$. Retinal blood flow measurement with laser speckle flowgraphy showed that the impaired retinal circulation response to both flicker light and systemic hyperoxia in db/db mice improved with $V_P$. Furthermore, the prolonged implicit time of b-wave and oscillatory potentials in electroretinography was prevented, and immunohistochemical analysis showed reduced microglial activation, gliosis, and vascular leakage. The enzyme-linked immunosorbent spot assay confirmed vaccinated mice had no auto-immune response against prorenin itself. The present data suggest that vaccination against prorenin is an effective and safe measure against the early pathological changes of diabetic retinopathy in T2D.

## Introduction

Prorenin is the most upstream protein in the renin-angiotensin system (RAS). Until the early 2000s, prorenin was viewed as an inactive form of renin; however, the findings of two studies on prorenin published in 2003 and 2004 [1, 2] greatly changed the situation. The first study found that prorenin binds to the (pro)renin receptor via a part of the prosegment called the handle region (HR), not via the renin region [1], and the second identified the mechanism by

**Funding:** This research was funded by the Ministry of Education, Culture, Sports, Science and Technology of Japan (grant numbers 18H06266 and 20K09835 [to H.Y.]).

**Competing interests:** The authors have declared that no competing interests exist.

which prorenin and the (pro)renin receptor activate the local RAS [2]. The binding of prorenin to the (pro)renin receptor leads to non-proteolytic activation of prorenin, which is followed by production of angiotensin II without conversion into mature renin, and direct activation of the receptor, which is followed by intracellular signaling by a mitogen-activated protein kinase, such as extracellular signal-regulated kinase (ERK) and p38 [3].

Levels of prorenin and (pro)renin receptor were reported to be closely correlated with the presence and severity of diabetic retinopathy [4–6]. Subsequently, a decoy peptide based on the sequence of the HR was used as a prorenin blocker and was found to have a beneficial, inhibitory effect on diabetic retinopathy by mitigating the production of vascular endothelial growth factor (VEGF) and intercellular adhesion molecule-1 (ICAM-1) [6–9]. Interestingly, some studies in diabetic rodent models showed that blocking prorenin had a more beneficial effect than inhibiting angiotensin II [7, 10], suggesting that conventional angiotensin II blockade is insufficient for preventing diabetic retinopathy. Therefore, a prorenin blocker is expected to become a new treatment for diabetic retinopathy. However, to date no prorenin blocker is available for clinical use.

Traditionally, vaccination has been used to prevent infectious diseases. Nowadays, common non-infectious diseases, such as hypertension, stroke, and Alzheimer's disease, are becoming a target for vaccine therapy in the form of self-antigens [11–16]. After successful vaccination, the therapeutic or preventive effect can last for a long time and may have several benefits compared with conventional treatment, such as lower cost and no need for daily medication. In fact, vaccine therapy targeting angiotensin II has been extensively studied and shown to have beneficial effects, such as reducing blood pressure [17, 18], improving cardiac function [19, 20], and protecting the brain from infarction [21].

Type 2 diabetes (T2D) is highly prevalent and frequently leads to diabetic retinopathy. The peptide vaccine approach may enable us to create a new treatment for diabetic retinopathy by focusing on the pivotal role of prorenin and (pro)renin receptor in the pathogenesis of this disease. Therefore, in this study we aimed to develop a prorenin peptide vaccine by the peptide vaccine approach [11, 22]. To do so, we tested three different antigens from the prorenin prosegment to develop a prorenin peptide vaccine. Then, we assessed the efficacy of this vaccine in the retina of db/db mice as a model of T2D.

## Materials and methods

### Animals

One week before the experiments, we purchased the following 7-week-old male mice from Charles River Laboratories (Wilmington, MA, USA): C57BL/6J for antigen determination (n = 24), C57BL/KsJ-db/db (BKS.Cg- +Lepr$^{db}$/+Lepr$^{db}$/J) as the T2D model (n = 18), and db/m (BKS.Cg- DOCK7$^m$+ +Lepr$^{db}$/J) as the non-diabetic control (n = 12). The total number of mice that we currently used was fifty-four. Mice were housed in a room at a controlled temperature with a 12-hour dark/12-hour light cycle with free access to food and water. The number of animals in each group was determined to be six according to our pilot study and previous works elsewhere [15, 21]. Three mice were housed in one cage. All procedures were undertaken under systemic anesthesia with 3% isoflurane. After finishing all the procedures, animals were euthanized with $CO_2$ inhalation and subsequent cervical dislocation. All efforts were made to minimize suffering and maintain the environment. The animal experiments were approved by the Ethical Committee of Nihon University and performed in accordance with the tenets of the Association for Research in Vision and Ophthalmology (ARVO) and the Animal Research: Reporting of In Vivo Experiments (ARRIVE).

To minimize potential confounders such as the order of treatments and measurements, the animals were divided into a half in each group and the procedures were performed in the first half of all the groups and the last of all the groups. There was no exclusion criterion in the current experiment.

## Vaccine preparation

The tertiary structure of murine prorenin (PDB ID: 5MKT) was depicted by using graphical imaging software CueMol2 (Molecular Visualization Framework; http://www.cuemol.org/) (Fig 1A). Prorenin consists of renin (white ribbons) and the prosegment (colored ribbon), which covers the enzymatic active site on renin [1]. From the 43 amino acids of prosegment, we selected three different amino acid sequences (Fig 1B): E1 (L$^{1P}$PTRTATFERIPLKKMP$^{17P}$) and E2 (T$^{7P}$FERIPLKKMP$^{17P}$), both of which contains the HR I$^{11P}$PLKK$^{15P}$, and E3 (M$^{16P}$PSVREILEER$^{26P}$), which does not contain the HR. E1, E2, and E3 were conjugated to keyhole limpet hemocyanin (KLH) [11]. For each mouse, we prepared 100 μl of peptide vaccine solution, which contained 20 μg of peptide vaccine diluted in 50 μl of normal saline and 50 μl of adjuvant, as described below. We used Freund's complete adjuvant (Wako Pure Chemical Industries, Ltd., Osaka, Japan) in the first vaccine and Freund's incomplete adjuvant (Wako Pure Chemical Industries, Ltd.) in the subsequent one [15, 16, 20, 21].

## Vaccination of C57BL/6 mice to determine the antigen for prorenin peptide vaccine

To determine which amino acid sequence from the prosegment elicited the best immune response, we injected C57BL/6J mice with E1 conjugated to KLH (E1-KLH), E2-KLH, and E3-KLH, as well as unconjugated KLH as a control vaccine. The first vaccination was performed at age 8 weeks and the second at age 10 weeks [15, 16, 20, 21] (Fig 1C). The injection site was the nape of the neck. Every 2 weeks until 10 weeks after the first vaccination, 10 μL of blood was collected from the tail veins. The samples were kept at -80°C until measurement of antibody titers.

## Vaccination to test the effect of prorenin peptide vaccine in db/db mice as a T2D model

Diabetic db/db and non-diabetic db/m mice were randomly immunized at age 8, 10, and 17 weeks (Fig 3A), and blood sugar levels, body weight, and antibody titer were monitored every 2 weeks until 24 weeks of age. H.Y. was aware of the group allocation throughout the current experiment. Blood sugar levels were measured from the tail veins (TERUMO Co., Ltd., Tokyo, Japan). At 24 weeks of age, blood pressure was measured once by the tail-cuff method with a blood pressure monitor (THC31; Softron, Tokyo, Japan). After the measurement, all animals were euthanized and the eyeballs were extracted, as follows: sternotomy was performed under systemic anesthesia with 3% isoflurane; normal saline was perfused into the left ventricle to wash out the circulating blood, immediately followed by perfusion of 4% paraformaldehyde (PFA), and the eyeballs were enucleated. The eyeballs were kept in 4% PFA at 4°C overnight; then, after a couple of washings in phosphate-buffered saline (PBS), they were embedded in Tissue-Tek OCT Compound (Sakura Finetek Japan, Tokyo, Japan) and stored at -80°C until analysis.

## Measurement of retinal blood flow

Retinal blood flow measurement was performed with a laser speckle flowgraphy (LSFG)-micro system (Softcare Co., Ltd., Fukutsu, Japan) [23]. The principle of LSFG-micro is the same as

**A**

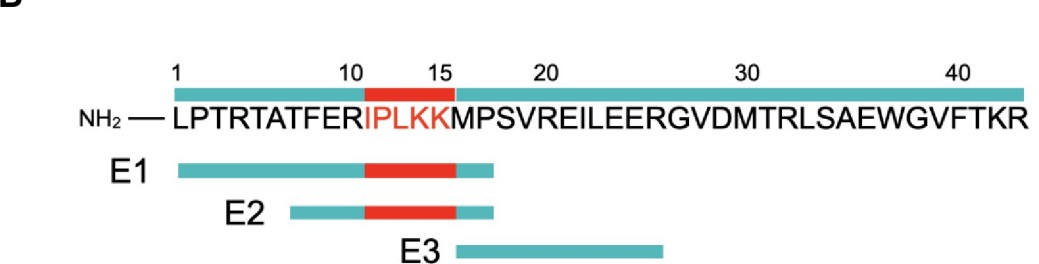

**B**

**C**

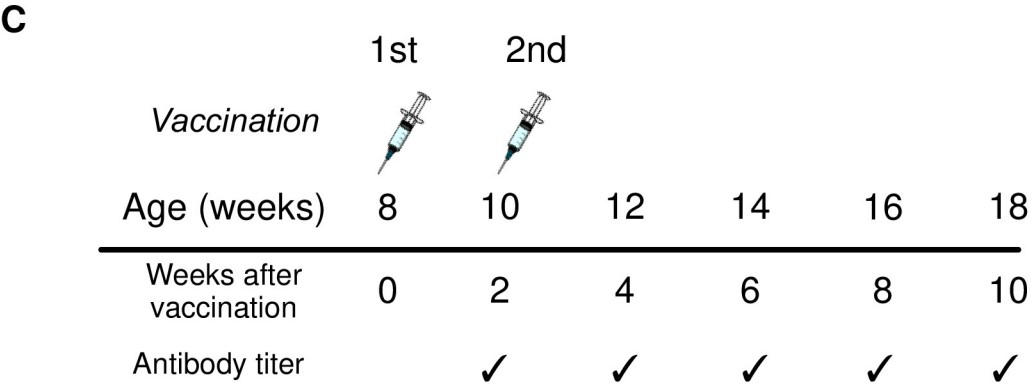

**Fig 1. Planning of optimization of prorenin peptide vaccine.** (A) Tertiary structure of prorenin. Prorenin consists of the prosegment (blue and red) and mature renin (white). (B) Antigen E1 and E2 contained the handle region, but E3 did not. (C) Schedule for vaccination and antibody titer measurement.

that of conventional LSFG, which has been used to non-invasively measure ocular circulation in humans with diabetic retinopathy [24, 25]. In LSFG, the blurring of the speckle pattern generated from moving blood cells is represented as the mean blur rate (MBR), which is recognized

as a relative index of blood velocity. We measured the MBR on the optic nerve head (ONH) to evaluate the changes of total retinal circulation after the stimuli described below.

## Induction of hyperoxia

Systemic hyperoxia was induced by inhalation of 100% oxygen for 10 minutes, as described in our previous study [23]. Briefly, the mean of three flow measurements obtained at 1-minute intervals for 3 minutes served as the baseline value before initiation of hyperoxia. Retinal blood flow was measured every minute for a total of 20 minutes, i.e., during the 10-minute inhalation (hyperoxia) and for 10 minutes afterwards (normoxia).

## Flicker light stimulation

The frequency of flicker light stimulation was set at 12 Hz on the basis of our recent results [23]. Before induction of flicker stimuli, ambient light was reduced to 1 lux or less. The mice were dark-adapted for 2 hours, and the light intensity for flicker light stimulation was set at 30 lux for the rod-dominant mouse retina. Before initiating flicker light stimulation, the mean of three measurements obtained at 20-second intervals over 1 minute was calculated as the baseline value. Then, retinal blood flow was measured at 20-second intervals during 3 minutes' flicker stimulation and for 3 minutes thereafter.

## Electroretinogram in db/db mice

Mice were dark-adapted for at least 6 hours before an electroretinogram (ERG) was performed. For ERG, the mice were transferred to a room with dim red light. The pupils were dilated with 0.4% tropicamide and a full-field ERG was recorded with PuREC (Mayo Corporation, Inazawa, Japan) under inhalation anesthesia (2% isoflurane). A ground electrode was placed at the tail, and a reference electrode was put in the mouth. Corneal electrodes were attached to the surface of the cornea. To gain the maximum response of both cones and rods, 3.0 candela.s/m$^2$ of flash was used. The implicit times of the a- and b-waves were automatically measured by identifying the maximum negative and positive peaks of the trace of the ERG. The measurement was repeated three times, and the mean value was calculated. Oscillatory potentials (OPs) were isolated by setting the high-pass digital filter at 75 Hz. We selected OP1, OP2, and OP3 and calculated the total sum of the OPs (ΣOPs).

## Antibody titer measurement

Enzyme-linked immunosorbent assay (ELISA) was performed to measure the antibody titer after vaccination [15, 16, 20, 21]. Briefly, a 96-well plate was coated with bovine serum albumin (BSA)-conjugated epitope (Peptide Institute Inc., Ibaraki, Japan) at 10 μg/mL and incubated overnight at 4°C. On the next day, to avoid non-specific binding, the plate was blocked for at least 2 hours with 5% skim milk in PBS containing 0.05% Tween 20. Diluted sera (from 10- to 32,500-fold) were applied to the wells and incubated overnight at 4°C. Each well was washed with PBS containing 0.05% Tween 20 and incubated with horseradish peroxidase-conjugated anti-mouse IgG antibody (GE Healthcare, Chicago, IL, USA) for at least 3 hours at room temperature. After washing, wells were incubated with the chromogenic substrate 3,3'-5,5'-tetramethyl benzidine (Sigma-Aldrich, St. Louis, MO, USA) for 30 minutes. Absorbance at 450 nm was measured with a microplate reader (Bio-Rad Laboratories, Inc., Hercules, CA, USA) after color development was stopped by 0.5 N sulfuric acid. The antibody titer of each sample was determined from the serum dilution that showed half the maximum absorbance of the plate reader by using GraphPad Prism 6 software (GraphPad Software, Inc., La Jolla, CA, USA).

## ELISA for measurement of prorenin plus renin and angiotensin I

The concentration of prorenin plus renin and angiotensin I in blood was measured by using an ELISA kit for prorenin plus renin (Lifespan Biosciences, Inc., Seattle, WA, USA) and for angiotensin I (SPI-Bio, Montigny Le Bretonneux, France). All procedures were performed according to the manufacturer's instructions.

## Western blotting

One microgram of recombinant prorenin (AnaSpec, Inc., Fremont, CA, USA) and renin (AnaSpec, Inc.) and 2 μg of BSA-conjugated E2 antigen were electrophoresed by 10% sodium dodecyl sulfate polyacrylamide gel electrophoresis and blotted onto polyvinylidene difluoride membrane (Millipore, Bedford, MA, USA). The blotted membranes were incubated with sera from mice immunized with E2-KLH or KLH or with commercially available anti-prorenin/renin antibody (dilution, 1:500; LifeSpan BioSciences, Inc.). After subsequent incubation with horseradish peroxidase-conjugated secondary antibody, immunoreactivity was detected with the enhanced chemiluminescence system (GE Healthcare).

## Immunohistochemistry

Ten micrometer-thick sections were made with a cryostat (HM505, Microm, Walldorf, Germany). The sections were stained with isolectin B4 (IB4) (dilution, 1:400; Thermo Fisher Scientific, Waltham, MA, USA) and rabbit polyclonal antibody against (pro)renin receptor (dilution, 1:200; Sigma-Aldrich, St. Louis, MO, USA), glial fibrillary acidic protein (GFAP; ready to use, Dako, Glostrup, Denmark), ionized calcium-binding adapter molecule 1 (iba-1; dilution, 1:200; Wako Pure Chemical Industries Ltd., Osaka, Japan), and vascular endothelial growth factor (VEGF; 1:100, Sigma-Aldrich) overnight at 4°C and then incubated with secondary donkey anti-rabbit IgG (H+L) Alexa Fluor 488 (dilution, 1:400; Thermo Fisher Scientific) for 2 hours at room temperature. Albumin was immunohistochemically stained with the primary antibody (dilution, 1:200; Abcam, Cambridge, MA, USA) and incubated with an Alexa Fluor 594-conjugated anti-sheep IgG antibody (dilution, 1:400; Thermo Fisher Scientific). Immunofluorescent images were obtained by a FluoView 1000 confocal microscope (Olympus, Tokyo, Japan) and BZ-9000 microscope (Keyence, Osaka, Japan). To examine the expression of phosphorylated extracellular signal-regulated kinase (p-ERK) in the retina, the sections were stained overnight at 4°C with p-ERK (dilution, 1:100). Immunostaining was developed with a Histofine Simple Stain PO (M) Kit (Nichirei, Tokyo, Japan) in accordance with the instruction manual.

## Enzyme-linked immunosorbent spot assay

Enzyme-linked immunosorbent spot (ELISPOT) assay was performed as previously described [15, 16]. Briefly, 96 well plates with polyvinylidene fluoride (PVDF) (Millipore) were coated with interleukin 4 (IL-4) or interferon-gamma (IFN-γ) capture antibodies (R&D Systems, Minneapolis, MN, USA) overnight at 4°C. On the next day, splenocytes were extracted from control mice or immunized mice with prorenin vaccine by using two doses of Freund's adjuvant (first dose, Freund's adjuvant complete; second dose, Freund's adjuvant incomplete; Sigma-Aldrich) and were incubated with recombinant mouse prorenin 1 μg/ml (Abcam), KLH 10 μg/ml (ENZO Life Sciences, Farmingdale, NY, USA), or phorbol myristate acetate (PMA) 0.1 μg/ml and ionomycin 0.1 μg/ml (Sigma-Aldrich) at 37°C for 45 hours. After incubation with streptavidin-alkaline phosphatase, spots were developed by 5-Bromo-4-chloro-3' indolylphosphate p-toluidine salt (BCIP) and nitro blue tetrazolium chloride (NBT) (R&D Systems). The spots were counted with a microscope (BZ-X810, Keyence).

## Statistical analysis

The results are presented as means ± standard error (SE). The data were analyzed with Eku-seru-Toukei 2010 (Social Survey Research Information Co., Ltd., Tokyo, Japan). Differences between groups were analyzed by an unpaired *t* test. Multiple comparison analysis was performed by one-way ANOVA followed by post hoc analysis (Tukey's test). A P value less than 0.05 was considered statistically significant.

# Results

## Determination of the antigen for developing a prorenin peptide vaccine

At 2 weeks postimmunization, the antibody titer was significantly increased in the E1-KLH and E2-KLH groups, but not in the E3-KLH group. The titer reached the maximum level at 4 weeks postimmunization (Fig 2A). The antibody titer of E2-KLH remained significantly higher than that of E1-KLH until 10 weeks postimmunization. To evaluate the specific reactivity of prorenin peptide vaccine with the full length of prorenin, we performed immunoblotting analysis by using sera obtained from the mice immunized with E2-KLH (Fig 2B). These sera bound to recombinant murine prorenin and BSA-conjugated E2 but not to recombinant renin. Anti-prorenin/renin antibody clearly showed blotting bands in both recombinant prorenin and renin but not in BSA-conjugated E2. Therefore, E2 was determined to be the best antigen for use as a prorenin peptide vaccine ($V_P$) in the current study and was used in the subsequent experiments.

## Antibody titer, body weight, blood glucose, blood pressure, renin activity, and (pro)renin receptor expression in db/m and db/db mice after injection of $V_P$

Immunization was performed in 8-, 10-, and 17-week-old db/db and db/m mice, as shown in Fig 3A. $V_P$ successfully raised the antibody titer in db/db mice (Fig 3B) but, compared with KLH, did not affect body weight or blood glucose levels of db/db mice (Fig 3C and 3D). Because the RAS plays a crucial role in maintaining blood pressure, we measured blood pressure once at age 24 weeks (Fig 3E); neither systolic nor diastolic blood pressure showed a difference between any of the groups. To examine the effect of $V_P$ on the concentration of prorenin and renin activity (renin and activated prorenin), we measured the serum concentration of prorenin plus renin and angiotensin I (Fig 3F): The concentration of prorenin plus renin was significantly higher in db/db + KLH (control vaccine) than in db/m + KLH and significantly lower in db/db + $V_P$ than in db/db + KLH. The angiotensin I concentration was significantly higher in db/db + KLH and db/db + $V_P$ than in db/m + KLH but was not different between db/db + KLH and db/db + $V_P$. In addition, immunohistochemical analysis showed no obvious difference between the groups in the expression of (pro)renin receptor in the retina.

## Protective effect of $V_P$ on glial function (retinal blood flow response to systemic hyperoxia and flicker light stimulation)

Systemic hyperoxia-induced decrease of retinal blood flow was confirmed in non-diabetic db/m + KLH, as previously reported [23]. In db/db + KLH, retinal blood flow was significantly increased. In contrast, in db/db + $V_P$, the response of retinal blood to hyperoxia was almost comparable to that in db/m + KLH. To further clarify the effect of $V_P$ on retinal blood flow response to stimuli, we used flicker stimulation (Fig 4B). Flicker light stimulation was reported to induce a gradual increase of retinal blood flow to meet tissue

**A**

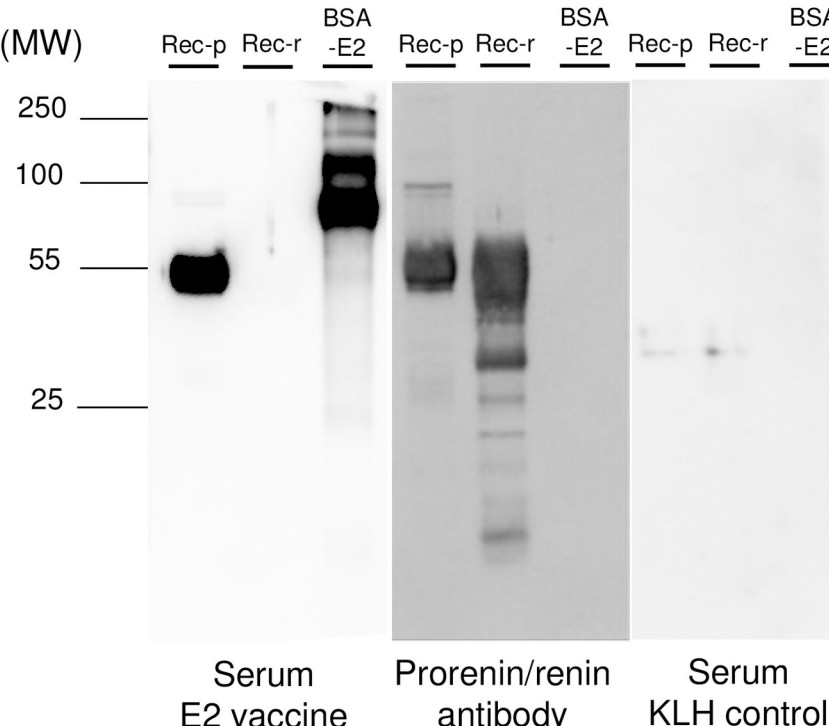

**B**

**Fig 2.** (A) Antibody titer in mice inoculated with E1 conjugated to KLH(E1-KLH), E2-KLH, E3-KLH, or KLH (unconjugated KLH as control) (n = 6 in each) (mean ± SE) measured by ELISA. $^*p < 0.05$ vs E1. (B) Confirmation of specific binding of E2 to prorenin. Sera were collected from mice inoculated with E2-KLH or KLH. A commercially available anti-prorenin/renin antibody was used as a positive control. BSA, bovine serum albumin; ELISA, enzyme-linked immunosorbent assay; KLH, keyhole limpet hemocyanin; MW, molecular weight; Rec-p, recombinant prorenin; Rec-r, recombinant renin.

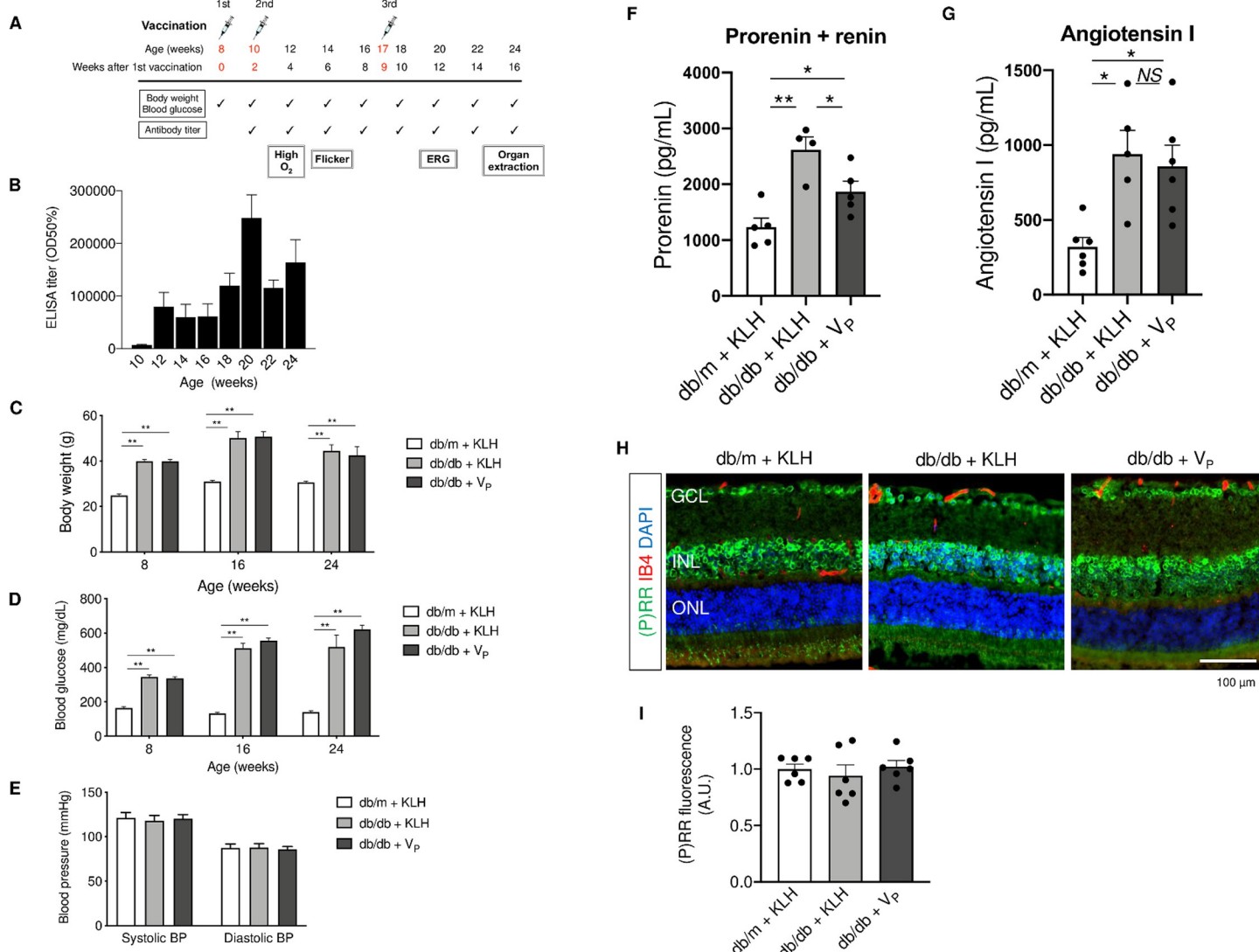

**Fig 3. Immunization with a prorenin peptide vaccine.** (A) Vaccination schedule and experimental plan in diabetic db/db and non-diabetic control db/m mice. The body weight and blood glucose levels were monitored every 2 weeks. Antibody titer measurement was started from 2 weeks after the first vaccination and continued every 2 weeks. Hyperoxia ("High O₂" in the figure), flicker light stimulation ("Flicker"), and electroretinography ("ERG") were performed at 12, 14, and 20 weeks of age, respectively, and eyeballs were surgically removed ("Organ extraction") at week 24. (B) Antibody titer (mean ± SE) in db/db + V$_P$ mice. (C) Body weight (mean ± SE) in each group at 8, 16, and 24 weeks of age. (D) Blood glucose levels (mean ± SE) in each group at 8, 16, and 24 weeks of age. (E) Blood pressure (mean ± SE) in each group 16 weeks after the first vaccination. (F) Prorenin plus renin concentration in the blood at 24 weeks of age. (G) Angiotensin I concentration in the blood at 24 weeks of age. (H) and (I) Expression of (pro)renin receptor [(P)RR] (green) in the retina at 24 weeks of age. Retinal vessels (red) were visualized by IB4. BP, blood pressure; DAPI, 4′,6-diamidino-2-phenylindole; ELISA, enzyme-linked immunosorbent assay; GCL, ganglion cell layer; IB4, isolectin B4; INL, inner nuclear layer; KLH, keyhole limpet hemocyanin; ONL, outer nuclear layer; P(RR), (pro)renin receptor; V$_P$, prorenin peptide vaccine. Scale bar = 100 μm. $^*p < 0.05$, $^{**}p < 0.01$, NS = not significant. N = 6 in each group except for the measurement of the prorenin plus renin and angiotensin I concentration.

demand [23], and we observed an increase of retinal blood flow in db/m + KLH. However, retinal blood flow showed an abnormal response in db/db + KLH in that it gradually decreased compared with baseline. In db/db + V$_P$, retinal blood flow response to flicker light stimulation gradually increased, similar to the normal response seen in db/m + KLH. Immunohistochemical analysis showed that V$_P$ ameliorated T2D-induced over-expression of GFAP (Fig 5A and 5B) and decreased the number of iba-1–positive microglia (Fig 5C and 5D).

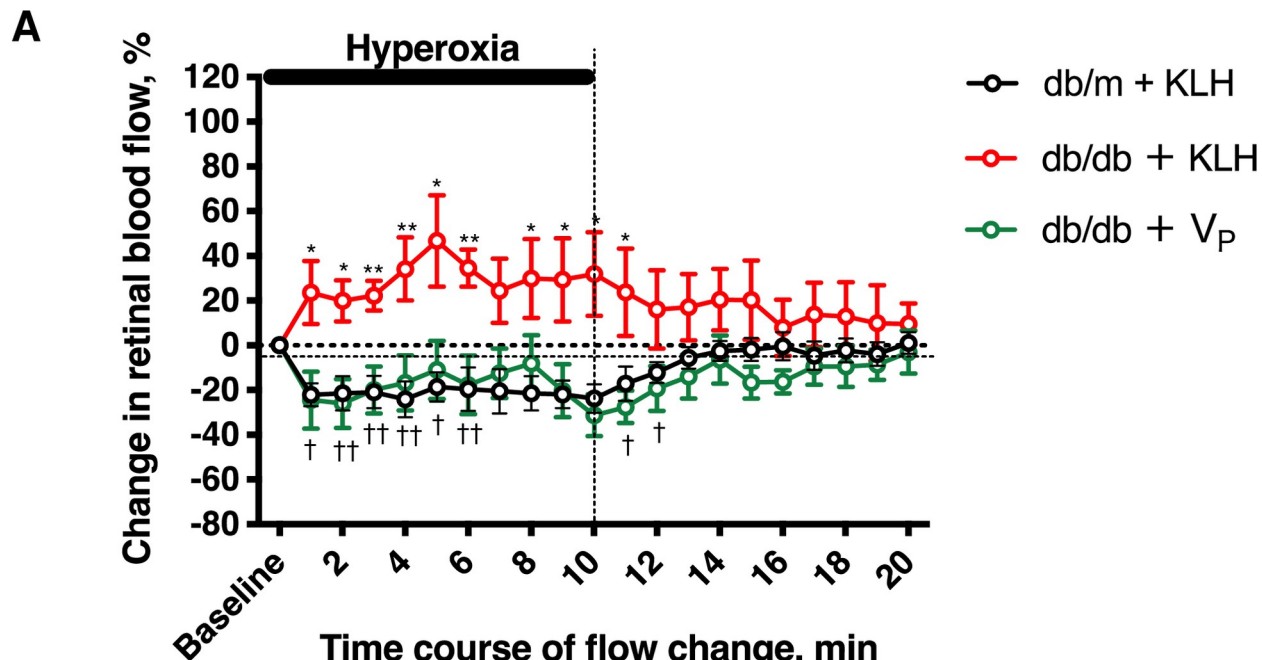

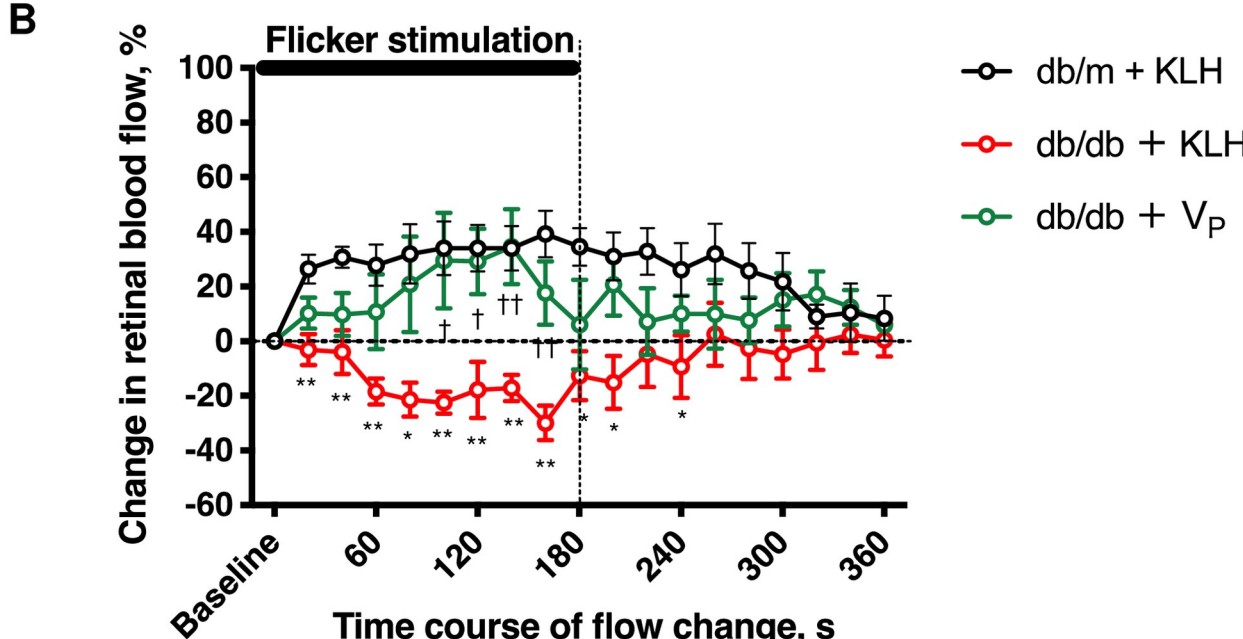

**Fig 4. Effect of prorenin peptide vaccine ($V_P$) on glial dysfunction.** (A) Effect of $V_P$ on retinal blood flow response to hyperoxia stimulation at 12 weeks of age. (B) Effect of $V_P$ on retinal blood flow response to flicker light stimulation at 14 weeks of age. KLH, keyhole limpet hemocyanin; $V_P$, prorenin peptide vaccine. $^* p < 0.05$, $^{**} p < 0.01$ vs. db/m + KLH, $^† p < 0.05$, $^{††} p < 0.01$ vs. db/db + KLH, NS = not significant. N = 6 in each group.

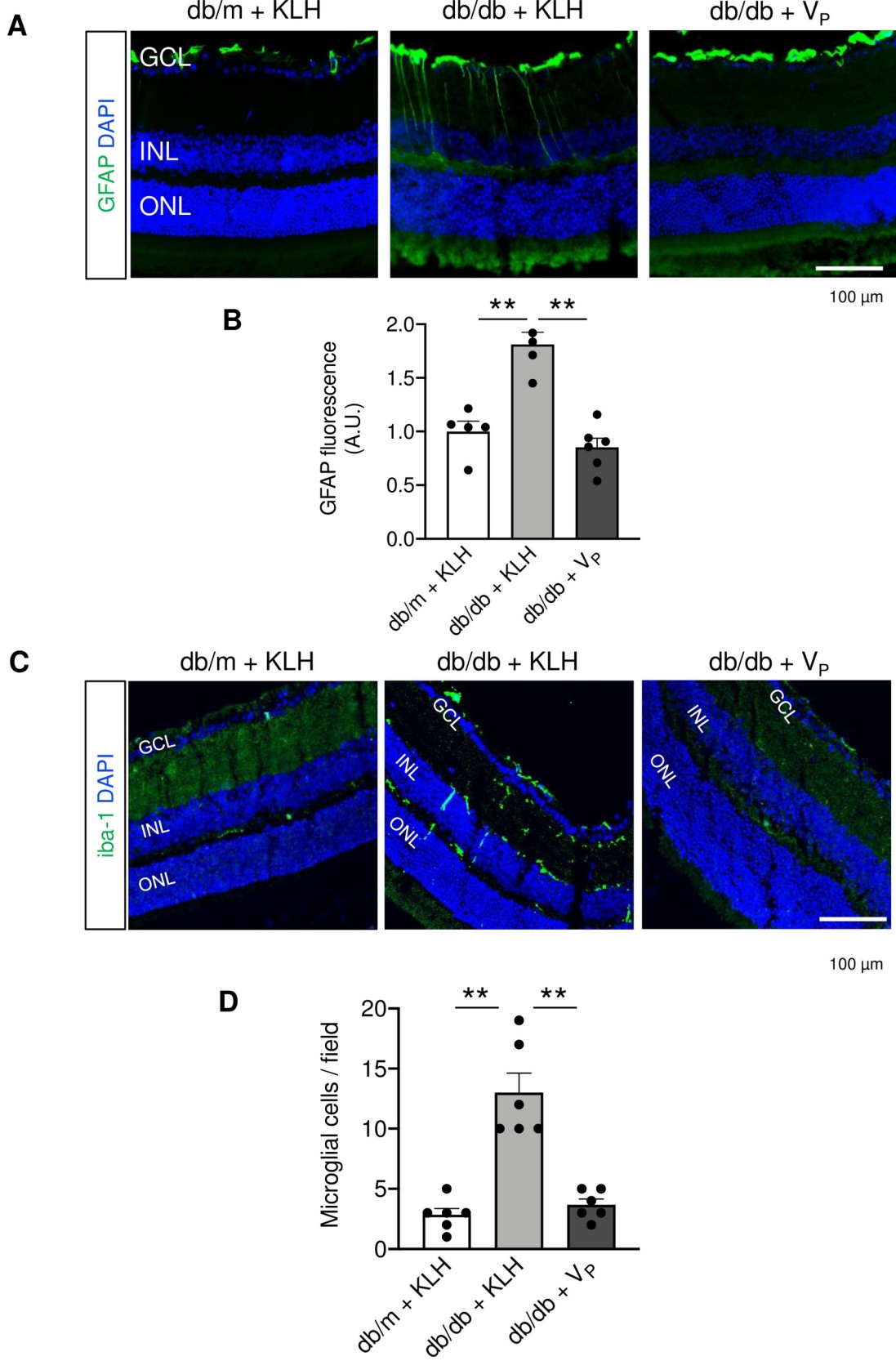

**Fig 5. Immunohistochemical analysis of gliosis and microglia activation.** (A) and (B) Gliosis was evaluated by immunofluorescence of glial fibrillary acidic protein (GFAP) at 24 weeks of age. (C) and (D) Microglia activation was quantified by ionized calcium-binding adapter molecule 1-positive cells at 24 weeks of age. DAPI, 4′,6-diamidino-2-phenylindole; GCL, ganglion cell layer; GFAP, glial fibrillary acidic protein; iba-1, ionized calcium-binding adapter molecule 1; INL, inner nuclear layer; KLH, keyhole limpet hemocyanin; ONL, outer nuclear layer; $V_P$, prorenin peptide vaccine. Scale bar = 100 μm. $^{**}p < 0.01$. N = 6 in each group except for GFAP expression.

## Protective effect of $V_P$ on retinal neuronal function in db/db mice

Representative ERG traces are shown in Fig 6A. We found no significant difference in the amplitude or implicit time of the a-wave between the groups (Fig 6B) or in the amplitude of the b-wave (Fig 5C); however, the implicit time of the b-wave was significantly prolonged in

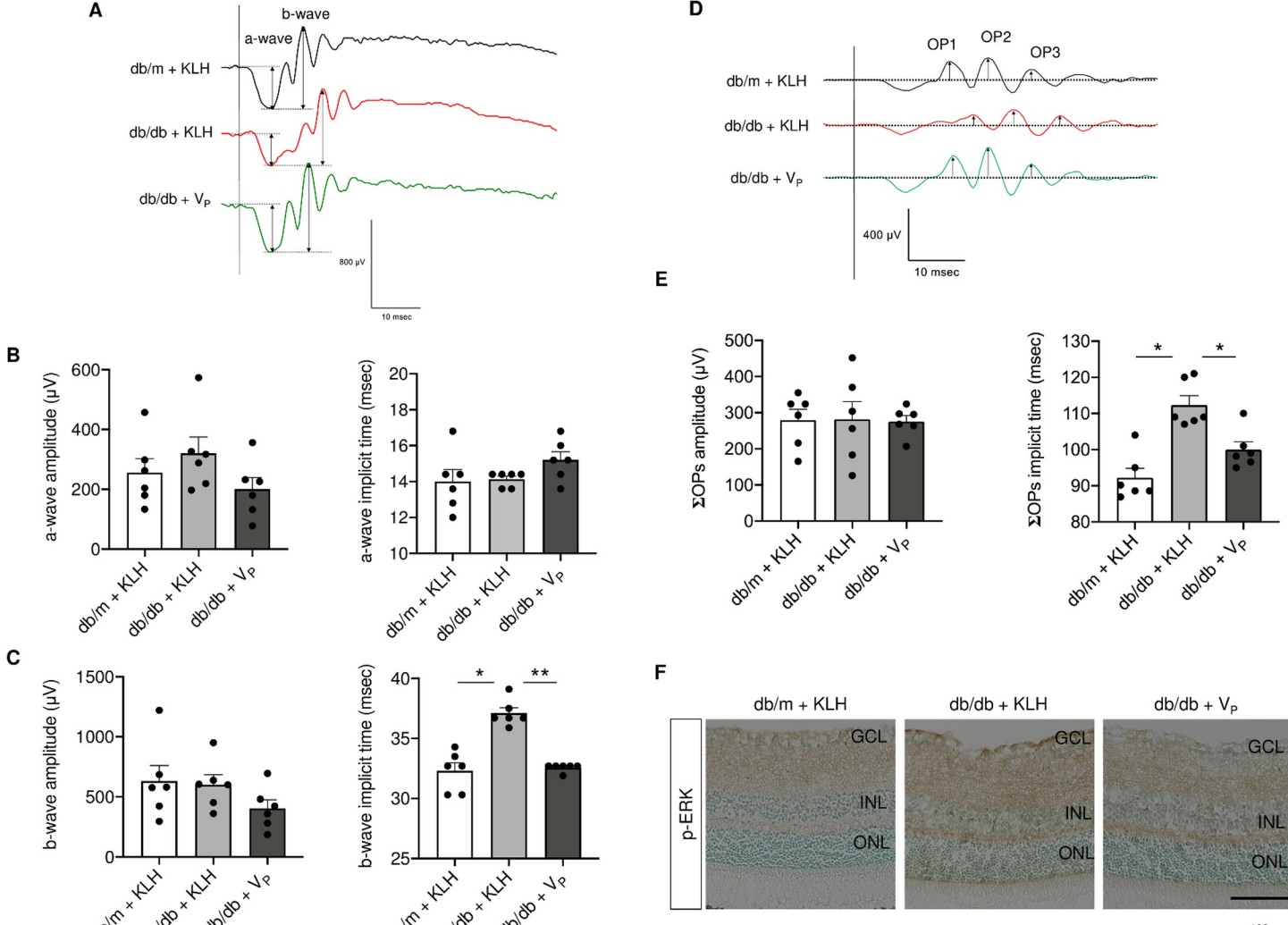

**Fig 6. Electroretinography at 20 weeks of age in diabetic (db/db) and non-diabetic (db/m) mice after injection of prorenin peptide vaccine ($V_P$).** (A) Representative traces in db/m + KLH (black line), db/db + KLH (red line), and db/db + $V_P$ (green line). (B) Quantitative analysis of a-wave amplitude and implicit time. (C) Quantitative analysis of b-wave amplitude and implicit time. (D) Representative traces of oscillatory potentials in db/m + KLH (black line), db/db + KLH (red line), and db/db + $V_P$ (green line). (E) The amplitude and implicit time of the summed oscillatory potentials (ΣOP). $^*p < 0.05$, $^{**}p < 0.01$. (F) Immunohistochemistry for p-ERK. GCL, ganglion cell layer; INL, inner nuclear layer; KLH, keyhole limpet hemocyanin; ONL, outer nuclear layer; OP, oscillatory potential; p-ERK, phosphorylated extracellular signal-regulated kinase; $V_P$, prorenin peptide vaccine. Scale bar = 100 μm. N = 6 in each group.

db/db + KLH compared with db/m + KLH. In db/db + $V_P$, the implicit time of the b-wave was almost normal, i.e., comparable to that in db/m + KLH (Fig 6C). To further evaluate the neuronal function of the inner layers of the retina after vaccination, we compared OPs between the groups (Fig 6D). Although we found no difference in the amplitude of $\Sigma$OPs, the implicit time of $\Sigma$OPs was significantly longer in db/db + KLH than in db/m + KLH but was significantly shorter in db/db + $V_P$ than in db/db + KLH (Fig 6E). In addition, db/m mice were immunized with $V_P$, which increased the antibody titer, as assessed by ELISA (S1 Fig). We found no prolongation of the implicit time of b-wave or $\Sigma$OPs in db/m + $V_P$ mice (S1 Fig). Immunohistochemical analysis of stress-induced mitogen-activated protein kinase expression showed a pronounced reduction of phosphorylated ERK in db/db + $V_P$ (Fig 6F).

### Inhibitory effect of $V_P$ on VEGF expression and vascular permeability

To further clarify the beneficial effect of $V_P$ on the pathogenesis of diabetic retinopathy, we performed immunohistochemical analysis to investigate whether $V_P$ ameliorates hyperpermeability in the retina of the T2D murine model. As shown in Fig 7, $V_P$ decreased the expression of VEGF and leakage of albumin in the retina of db/db + $V_P$ mice.

### T-cell activation after inoculation with $V_P$

To overcome concerns about the development of unfavorable autoimmune disorders related to vaccination, ELISPOT assay was performed with splenocytes isolated from E2-immunized mice. Stimulation with KLH significantly increased the number of IL-4 spots in the splenocytes (Fig 8A) and also the number of IFN-$\gamma$ spots (Fig 8B). Neither the IL-4 spots nor the IFN-$\gamma$ spots were increased by stimulation with prorenin recombinant. IgG subclass assay demonstrated that IgG1/IgG2a ratio was greater than 1.0 in db/m and db/db mice (S2 Fig). These data suggest that $V_P$ immunization activated T helper 2 (Th2) responses to induce antibody production and that intrinsic prorenin does not induce autoimmune activation.

### Comparison of $V_P$ with angiotensin II peptide vaccine ($V_A$)

To further elucidate the benefits of $V_P$ in the prevention of diabetic retinopathy, we studied the effects of angiotensin II peptide vaccine ($V_A$) in this diabetic model in the same manner as $V_P$. As shown in Fig 9A, $V_A$ led to an increase in the antibody titer; the increase was similar to that shown in previous reports [21]. The representative ERG traces are shown in Fig 9B. The implicit time of the b-wave and $\Sigma$OPs were significantly prolonged in db/db + $V_A$ compared with db/db + $V_P$ (Fig 9C).

## Discussion

In this study, we found that the HR is a crucial sequence for designing a prorenin peptide vaccine. Prorenin peptide vaccine successfully protected the function of Müller glia and neurons in the retina of the T2D model. Moreover, the vaccine prevented T2D-induced microglial activation and hyperpermeability of the retina. To our knowledge, this is the first study to demonstrate that vaccination may be an alternative approach to prevent new onset of diabetic retinopathy.

Prorenin consists of renin and the prosegment. The HR is a 5-amino acid sequence located in a 43-amino acid sequence of the prosegment [1]. Because binding of the HR to the (pro) renin receptor is the rate-limiting step of proteolytic activation of prorenin [1], decoy peptides were used as prorenin blockers, such as NH2-R$^{10P}$ ILLKKMPSV$^{19P}$-COOH in rats [2] and NH2-I$^{11P}$ PLKKMPS$^{18P}$-COOH in mice [7, 26]. In the present study, for unknown reasons E3

**A**

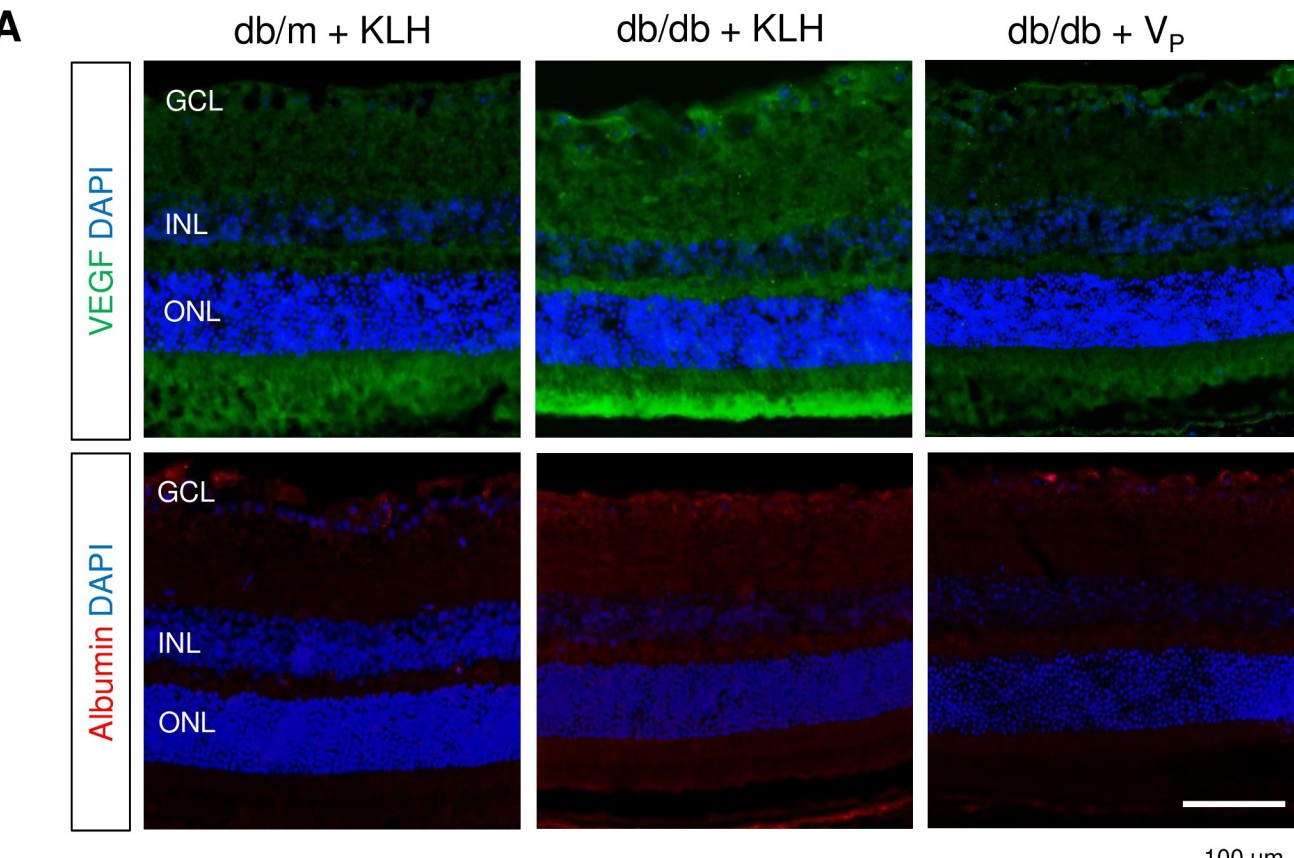

**B**

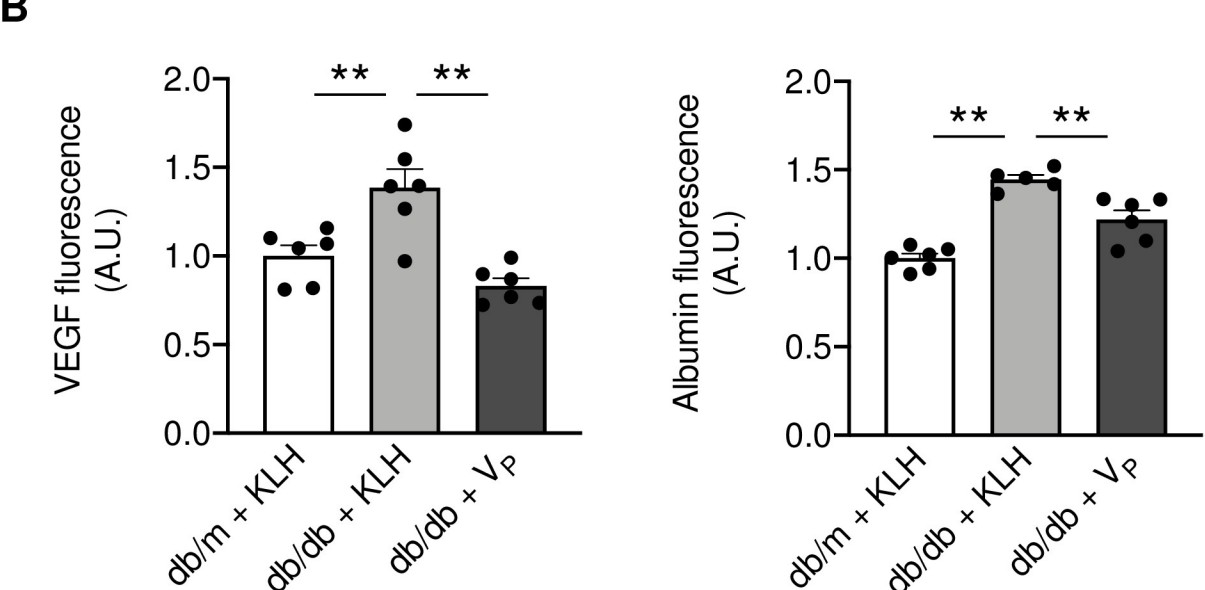

**Fig 7. Immunohistochemical analysis of vascular leakage in the retina of mice at 24 weeks of age.** (A) Vascular endothelial growth factor (VEGF; green), and albumin (red) for vascular leakage in the retina. (B) Quantification of VEGF expression and albumin in the retina. $^{**}p < 0.01$. DAPI, 4′,6-diamidino-2-phenylindole; GCL, ganglion cell layer; iba-1, ionized calcium-binding adapter molecule 1; INL, inner nuclear layer; KLH, keyhole limpet hemocyanin; ONL, outer nuclear layer; VEGF, vascular endothelial growth factor; $V_P$, prorenin peptide vaccine. Scale bar = 100 μm.

**A**

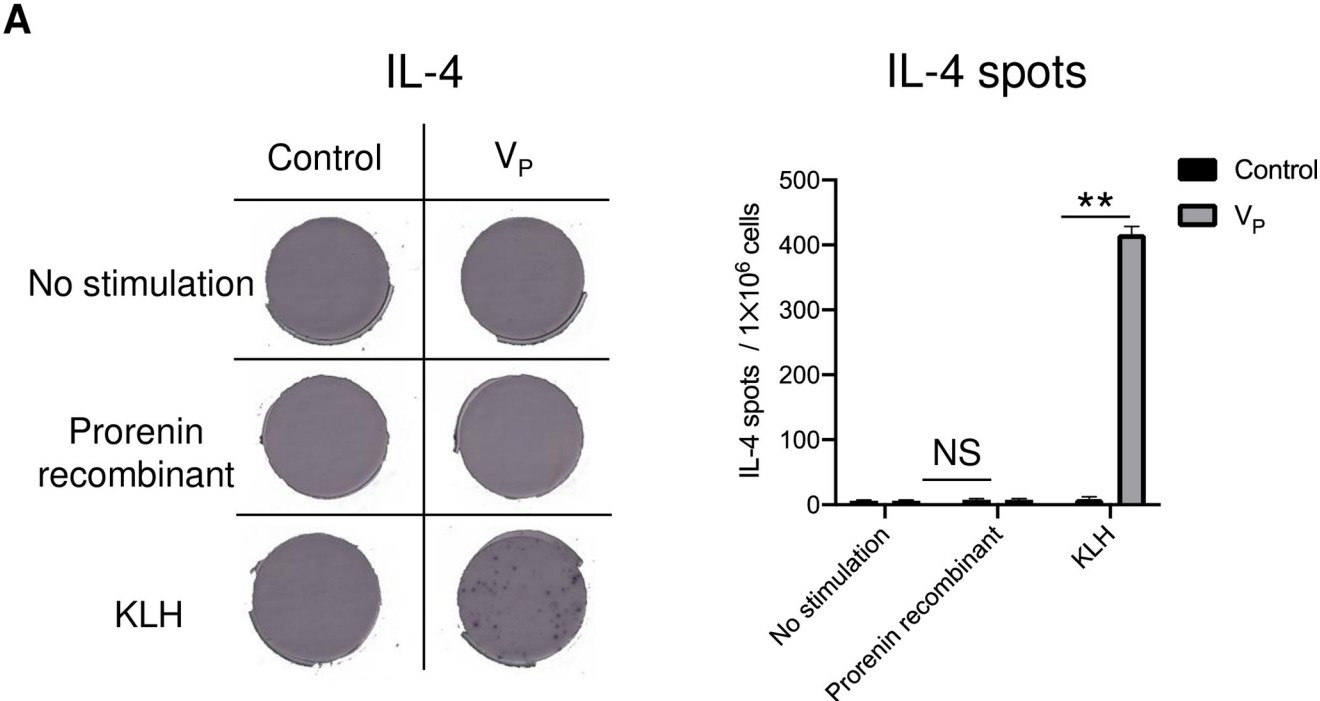

**B**

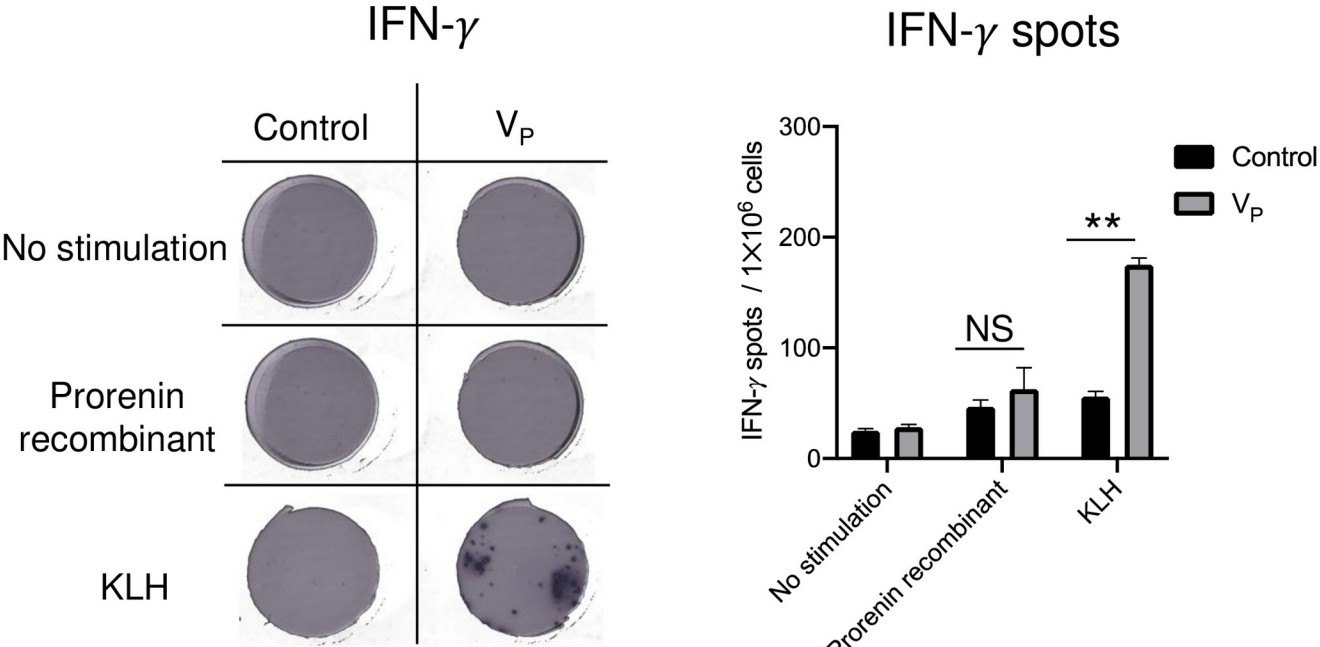

**Fig 8. Enzyme-linked immune absorbent spot assay with splenocytes ($1 \times 10^6$ cells) from C57BL/6 mice immunized with prorenin peptide vaccine ($V_P$).** The production of interleukin 4 (A) and interferon-gamma (B) was detected as black spots. The bar graphs show the number of spots per membrane. Control is the age-matched C57BL/6 mouse without vaccination. KLH, keyhole limpet hemocyanin; IFN-$\gamma$, interferon-gamma; IL-4, interleukin 4; $V_P$, prorenin peptide vaccine. $^{**}p < 0.01$.

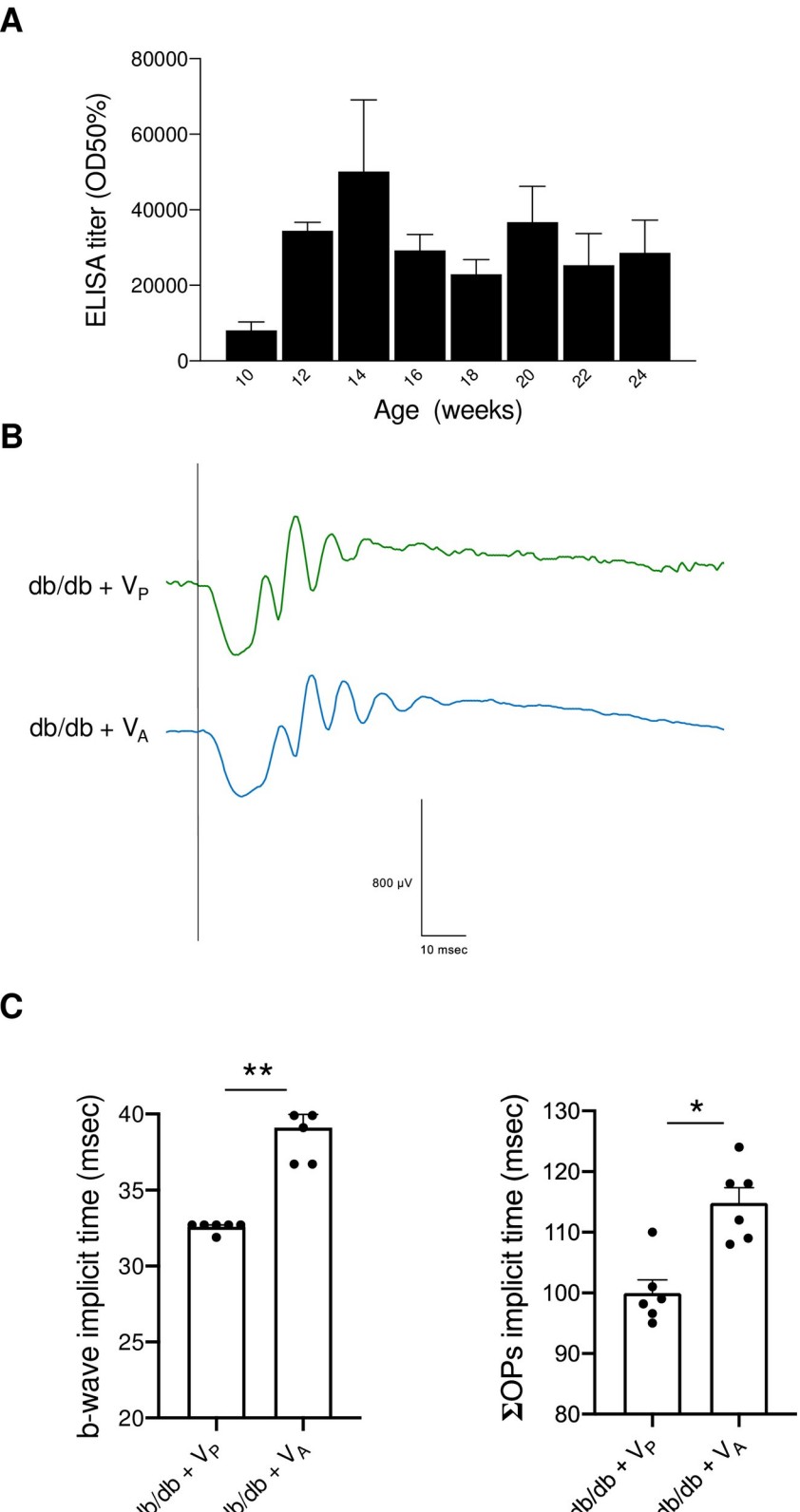

**Fig 9. Comparison of neuroprotection by angiotensin II peptide vaccine (V_A) and prorenin peptide vaccine (V_P) in the retina of db/db mice.** (A) Antibody titer (mean ± SE) in db/db + V_A mice. (B) Representative electroretinogram traces in db/db + V_P (green line) and db/db + V_A (blue line). (C) Implicit time of b-wave and total sum of the

oscillatory potentials in db/db + $V_P$ and db/db + $V_A$. $\Sigma$OPs, total sum of the oscillatory potentials; ELISA, enzyme-linked immunosorbent assay; OD50%, optical density at half-maximal binding; $V_A$, angiotensin II peptide vaccine; $V_P$, prorenin peptide vaccine. $^* p < 0.05$, $^{**} p < 0.01$.

($M^{16P}$PSVREILEER$^{26P}$) did not raise an antibody titer. Taken together, findings indicate that the HR ($I^{11P}$PLKK$^{15P}$) is a key sequence for creating a peptide vaccine against prorenin that efficiently inhibits the binding of prorenin to the (pro)renin receptor. In the present study, we aimed to design a vaccine that did not affect renin because of the crucial role in the RAS, and we confirmed that the vaccine-derived antibody specifically bound to recombinant prorenin but not to renin. The angiotensin I concentration, which is mainly dependent on renin activity, was not different between KLH and prorenin peptide vaccine. This result also indicates that the antibody does not interfere with renin activity in the circulation. Renin is a rate-limiting molecule in the generation of angiotensin II in the circulatory RAS [27]. Angiotensin II plays central roles in the physiological control of systemic blood pressure and sodium balance [28]. Hence, the prorenin peptide vaccine created in the current study will not affect the physical role of renin and will independently prevent tissue damage attributed to prorenin-activated RAS.

In the present study, we used db/db mice as a murine model of T2D. During the study, the body weight of the db/db mice steadily increased up to almost double that of the non-diabetic C57BL/6J mice. The amount of vaccine injected into each mouse was identical, despite their different body weights. In humans, a study found that the immune response of obese and diabetic individuals to vaccination was compromised compared with that of non-obese, non-diabetic individuals [29]. However, throughout the 16-week period of our study, the antibody titer of the overweight db/db mice remained almost comparable to that of the lean, non-diabetic C57BL/6 mice. The abovementioned study, which investigated whether obesity affects the antibody titer after influenza vaccine, clearly demonstrated that 12-month postvaccination titers were drastically lower in people with obesity [29]. In addition, a leptin-deficient, obese mouse (ob/ob) model was reported to show a significant decrease in the number of B cells in the bone marrow [30]. Thus, a further study is needed to longitudinally observe the postvaccination titer for more than 12 months in various models of T2D with obesity.

Because a vaccine affects the whole body, safety is prioritized when considering the clinical application. In the present study, the mice did not show any signs of severe side effects that led to differences in the size and weight of the control vaccine and prorenin peptide vaccine groups. Previously, the (pro)renin receptor was reported to be involved in impaired insulin sensitivity [31]. Our current study indicates that vaccination against prorenin does not improve blood glucose levels in T2D. For the first time, the present study found higher levels of prorenin in the circulation of T2D db/db mice. Furthermore, the study suggested that prorenin peptide vaccine does not affect the circulatory RAS, as mentioned above. Taken together, our results indicate that the antibody produced in response to prorenin peptide vaccine prevents prorenin from binding to the (pro)renin receptor but does not affect the circulatory RAS.

Recent studies have paid much attention to Müller glia by focusing on early pathological changes in the retina in T2D. Studies thereby showed that observing the retinal blood flow response to both systemic hyperoxia and flicker light stimulation is useful for examining glial function [32, 33]. Systemic hyperoxia induces vasoconstriction via a biophysiological effect of endothelin-1 released from glia [32]. In contrast, to meet the oxygen and nutrient demands of neurons, flicker light stimulation induces vasodilation via nitric oxide, which is mainly produced in glia [33]. The present study showed for the first time that T2D severely affects the

retinal blood flow response to hyperoxia and flicker light stimulation. Although the mechanism by which dysregulation of neurovascular coupling occurs in the retina in T2D is not fully understood, microglia may play a critical role [34]. Microglial activation is also one of the early events in the retina soon after T2D develops [35]. Prorenin was reported to directly activate cultured microglia, even when they were preincubated with an angiotensin II type 1 receptor blocker [36, 37], indicating that angiotensin II blockade is not sufficient to suppress microglial activation in diabetic retinopathy. We speculate that the antibody produced by vaccination against prorenin ameliorates microglial activation and subsequently reduces the expression of pro-inflammatory cytokines in the retina in T2D. Our data demonstrate for the first time that the cascade of prorenin-(pro)renin receptor activation was involved in glial dysfunction in the early pathogenesis of diabetic retinopathy.

Neurodegenerative change has come to be recognized as one of the early features of diabetic retinopathy [38–40]. In our ERG analysis, we found no significant reduction of the amplitudes of the a-wave, b-wave, or OPs in db/db diabetic mice; these findings are in contrast with a previous study [41]. However, in our study, the implicit time of the b-wave and OPs was significantly prolonged in the db/db diabetic mice. The reason for this discrepancy is unclear, but our data clearly showed the protective effect of prorenin peptide vaccine on the neuronal retina of diabetic mice. The effect of prorenin inhibition on the neurological function of the retina was not well defined, but a study with decoy prorenin inhibitor showed that, surprisingly, the decoy peptide resulted in a further worsening of neuronal function of the retina, as demonstrated by ERG and immunohistochemical analysis of p-ERK [42]. The authors of the study speculated that by binding to the (pro)renin receptor the decoy peptide may have subsequently stimulated intracellular signaling just beneath the receptor. On the other hand, the antibody produced by the current prorenin peptide vaccine is expected to prevent prorenin from binding to the (pro)renin receptor by covering the binding site located in the prosegment. In fact, the implicit time of b-wave and ΣOPs in non-diabetic db/m mice showed no difference between KLH and prorenin peptide vaccine (S1 Fig), indicating that prorenin peptide does not have any detrimental effect on neuronal function even in non-diabetic individuals. Taken together, the prorenin peptide vaccine was found to be safer and more effective in treating diabetic retinopathy than a peptide-based prorenin inhibitor.

In addition, binding of prorenin to the (pro)renin receptor leads to overexpression of VEGF in the retina of diabetic rodents independently of the angiotensin II pathway [7]. Thus, the current results also substantiate the beneficial effect of prorenin peptide vaccine in preventing the hyperpermeability that results in edema. In patients with T2D, diabetic macular edema severely affects central vision and is one of the major concerns in treating diabetic retinopathy. Therefore, prorenin peptide vaccine is expected to become an alternative or adjuvant therapy to anti-VEGF therapy in macular edema.

In 2003, a clinical trial of a vaccine for Alzheimer's disease was stopped because of a severe adverse effect, i.e., 3% of the participants experienced T cell-mediated aseptic meningoencephalitis [43]. Therefore, safety issues need to be carefully considered before taking the prorenin vaccine to the next step. The current ELISPOT assay clearly demonstrated that T-cells of the vaccine-immunized mice did not show an auto-immune response against prorenin. Furthermore, IgG subclass assay showed that the humoral immune response was significantly greater than the cellular immune response (S2 Fig). These results suggest that prorenin peptide vaccine is unlikely to cause prorenin-related autoimmune diseases.

Many studies have demonstrated beneficial protective effects of vaccination against angiotensin II in models of hypertension and brain ischemia. Moreover, angiotensin II has been suggested to be involved in the development of diabetic retinopathy [44–46]. In the present study, we administered three doses of angiotensin II peptide vaccine to db/db mice in the

same manner as prorenin peptide vaccine. The antibody titer of angiotensin II peptide vaccine was almost the same as the one found in a previous study [21]. However, angiotensin II peptide vaccine did not exhibit a protective effect, at least on neuronal function of the retina in T2D. Taken together, our data indicate that, in terms of vaccine therapy for diabetic retinopathy, prorenin is more suitable as a target molecule than angiotensin II.

The present study has several limitations. First, we did not determine the actual retinal levels of prorenin in the db/db mice. To our knowledge, no study has investigated the expression levels of prorenin in the retina of db/db mice. Besides the retina, one study clearly showed that mRNA levels of prorenin were increased in the kidney of db/db mice and that a decoy peptide suppressed the pathogenesis of diabetic nephropathy [26]. Another group reported that a decoy peptide successfully inhibited the effect of increased prorenin in streptozotocin-induced type 1 diabetes rodent model [7]. To further clarify whether prorenin peptide vaccine shows a beneficial effect locally or systemically, a future study needs to examine the expression and alteration of prorenin and angiotensin II in db/db mice with or without vaccination. Second, the mechanism by which prorenin peptide vaccine protects retinal neurovascular coupling and neuronal function in T2D is not yet determined. Prorenin was reported to be capable of activating microglia via the (pro)renin receptor, and chronic inflammation has been reported to take place prior to neuronal and vascular injuries [39, 47]. However, controversy remains regarding the involvement of the angiotensin II type 1 receptor. Some studies showed that conventional angiotensin II blockade inhibits microglial activation [48–51], but others found that angiotensin II is unnecessary for prorenin-induced activation of microglia [36, 52]. Presumably, microglia can be activated by either pathway. Third, we have not yet determined where the produced antibody is located in the db/db diabetic retina, i.e., whether the antibody remains within the blood vessels or leaks out of the blood vessels. To resolve this issue, we need to use a larger diabetic animal model, such as a pig, which will allow us to collect a large enough ocular sample [53] to determine the intraocular levels of the antibody.

## Conclusion

The current findings suggest that a prorenin peptide vaccine may be effective in preventing diabetic retinopathy by suppressing microglial activation, gliosis, neuronal dysfunction and hyperpermeability in T2D. When preparing a prorenin peptide vaccine, the HR must be included in the antigen sequence. To bring this technique to the next level, efforts must be made to establish a consensus about the efficacy and safety of peptide vaccine therapy for retinal diseases.

## Supporting information

**S1 Fig. The results of prorenin peptide vaccine ($V_P$) injection in non-diabetic db/m mice.** (A) Antibody titer of db/m mice immunized with $V_P$. (B) Electroretinography at 20 weeks of age in db/m mice immunized with keyhole limpet hemocyanin (KLH) control vaccine (db/m + KLH) and $V_P$ (db/m + $V_P$). (C) Implicit time of b-wave and total sum of the oscillatory potentials in db/m + KLH and db/m + $V_P$. ΣOPs, total sum of the oscillatory potentials; ELISA, enzyme-linked immunosorbent assay; KLH, keyhole limpet hemocyanin; OD50%, optical density at half-maximal binding; $V_P$, prorenin peptide vaccine.
(TIF)

**S2 Fig. IgG subclass analysis of the antibody of prorenin in db/m and db/db mice immunized with prorenin peptide vaccine.** Horseradish peroxidase-conjugated anti-mouse IgG1, IgG2a, IgG2b, and IgG2c (Abcam, Cambridge, MA, USA) were used to determine the IgG

subclasses. The IgG1/IgG2a ratio was greater than 1.0, showing that activated T helper 2 cells (Th2) predominated over T helper 1 cells (Th1) in both db/m and db/db mice.
(TIF)

**S1 Raw image.**
(TIF)

**S2 Raw image.**
(TIFF)

**S3 Raw image.**
(TIFF)

## Acknowledgments

We thank Akiko Tomioka and Yuka Yanagida for technical support.

## Author Contributions

**Conceptualization:** Harumasa Yokota, Hironori Nakagami, Taiji Nagaoka.

**Data curation:** Harumasa Yokota, Hiroki Hayashi.

**Formal analysis:** Junya Hanaguri.

**Funding acquisition:** Harumasa Yokota.

**Investigation:** Harumasa Yokota, Hiroki Hayashi.

**Methodology:** Harumasa Yokota, Hiroki Hayashi, Hironori Nakagami.

**Project administration:** Satoru Yamagami, Taiji Nagaoka.

**Resources:** Hironori Nakagami.

**Supervision:** Satoru Yamagami.

**Validation:** Junya Hanaguri.

**Visualization:** Harumasa Yokota, Hiroki Hayashi.

**Writing – original draft:** Harumasa Yokota, Hiroki Hayashi.

**Writing – review & editing:** Akifumi Kushiyama, Hironori Nakagami, Taiji Nagaoka.

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
