## [Decision Letter · Decision Letter 0]

25 Jun 2021

PONE-D-21-16317

Effect of prorenin peptide vaccine on the early phase of diabetic retinopathy in a murine model of type 2 diabetes

PLOS ONE

Dear Dr. Yokota,

Thank you for submitting your manuscript to PLOS ONE. After careful consideration, we feel that it has merit but does not fully meet PLOS ONE’s publication criteria as it currently stands. Therefore, we invite you to submit a revised version of the manuscript that addresses the points raised during the review process.

Please quantify your results for Figs. 3-6. Please evaluate the T cell response to immunization. Please add a control group for your in vivo efficacy study of the peptide vaccine.

We look forward to receiving your revised manuscript.

Kind regards,

Alfred S Lewin, Ph.D.

Academic Editor

PLOS ONE

Journal Requirements:

2. As part of your revisions, please provide additional details pertaining to your animal research procedures. Please revise your Methods to address the following points: (1) sample size: how many animals per group and how did you determine the numbers (power analysis? pilot study? previously published data/findings?); (2) provide complete information about all methods undertaken to minimize pain and distress of the animals in your work, including method of euthanasia which is not currently specified. (3) Please discuss your monitoring parameters (physical and behavioral signs to assess health and well-being), in addition to your humane endpoints (criteria used to determine when to euthanize animals in cases where animals become sick/moribund). (4) Please state the rate of mortality for animals who died unexpectedly (and state cause(s) of death). (6) Please also discuss supportive care that you provided to the animals. (7) Lastly, please complete and submit the ARRIVE Guidelines checklist (Essential 10) with your revision: https://arriveguidelines.org/resources/author-checklists

Reviewers' comments:

Reviewer's Responses to Questions

**Comments to the Author**

1. Is the manuscript technically sound, and do the data support the conclusions?

Reviewer #1: Partly

Reviewer #2: Yes

2. Has the statistical analysis been performed appropriately and rigorously? 

Reviewer #1: No

Reviewer #2: No

3. Have the authors made all data underlying the findings in their manuscript fully available?

Reviewer #1: Yes

Reviewer #2: Yes

4. Is the manuscript presented in an intelligible fashion and written in standard English?

Reviewer #1: No

Reviewer #2: Yes

5. Review Comments to the Author

Reviewer #1: The manuscript titled “Effect of prorenin peptide vaccine on the early phase of diabetic retinopathy in a murine model of type 2 diabetes” by Yokota et al. tested vaccines using three epitopes from the prosegment of prorenin conjugated to keyhole limpet hemocyanin (KLH), evaluated the efficacy of one of them in diabetic retina using db/db mice, and showed protective effects against the early pathological changes of diabetic retinopathy in this model. While this study is interesting and maybe significant, there are some major concerns as listed below.

• While authors showed that E2-KLH conjugate had highest anti-prorenin antibody titer, this peptide is 11 amino acid long and could potentially activate autoreactive T-cells. T-cell response should be evaluated. IgG subclass of peptide-induced antibodies and levels of complement system proteins should also be investigated.

• The experimental design for in vivo efficacy study of the peptide vaccine lacked the db/m + Vp treated control group.

• In all figures with immunostaining, there are no quantitative measurements and statistical analysis (Fig 3H, Fig 4C, Fig 5F, Fig 6).

• As elevated prorenin in the retina is thought to contribute pathogenesis of diabetic retinopathy, the levels of prorenin/renin and Ang I/II in the retina with and without treatments should be measured. This is important as authors showed that this peptide vaccine seemed to only block prorenin-induced pathway(s) independent of Ang II, without affecting circulating renin level and it is important to know whether the beneficial effects of this peptide vaccine is mediated via systemic effects or locally via blocking prorenin and its receptor activation. It would be also interesting to compare the efficacy of the prorenin peptide vaccine with classic RAS blockers such as ACE inhibitors and Ang II receptor blockers.

• In the Introduction and Discussion, there is insufficient discussion on related background literature, why prorenin is an ideal target molecule in the prevention of diabetic retinopathy, some important early reports on prorenin in diabetic retinopathy are not mentioned. There is insufficient discussion on data presented and potential mechanisms.

Reviewer #2: Yokota et al. “Effect of prorenin peptide vaccine on the early phase of diabetic retinopathy in a

murine model of type 2 diabetes”

This is an interesting manuscript because it demonstrates that vaccination may be an alternative approach to prevent diabetic retinopathy. Furthermore, the authors report that T2D severely affects the retinal blood flow response to hyperoxia and flicker light stimulation. Moreover, the efficacy of the prorenin-directed vaccine indicates that prorenin contributes to glial dysfunction and early pathogenesis of diabetic retinopathy. Finally, they speculate that microglial activation may compromise neurovascular coupling and thereby promote retinopathy.

Specific Points

1. While the text states that there is no difference in the prorenin receptor level, Fig 3H shows that there is less in the retinal from the vaccinated animal. Please quantify the results shown in Fig 3H.

2. Please modify Fig 4A and B to indicate the times at which there are statistically significant differences between the experimental groups.

3. Please quantify the results shown in Fig 4C.

4. Please quantify the results shown in Fig 5F.

5. Please quantify the results shown in Fig 6.

6. PLOS authors have the option to publish the peer review history of their article (what does this mean?). If published, this will include your full peer review and any attached files.

Reviewer #1: No

Reviewer #2: No

---

## [Author Response · Author response to Decision Letter 0]

3 Dec 2021

We thank all the reviewers for providing constructive comments and valuable suggestions for the manuscript. The reviewer’s comments are addressed point-by-point below.

Responses to Reviewer 1

Comment 1: The manuscript titled “Effect of prorenin peptide vaccine on the early phase of diabetic retinopathy in a murine model of type 2 diabetes” by Yokota et al. tested vaccines using three epitopes from the prosegment of prorenin conjugated to keyhole limpet hemocyanin (KLH), evaluated the efficacy of one of them in diabetic retina using db/db mice, and showed protective effects against the early pathological changes of diabetic retinopathy in this model. While this study is interesting and maybe significant, there are some major concerns as listed below.

Response 1: Thank you very much for your positive remarks and useful suggestions. We have provided point-by-point responses to each of your comments below. 

Comment 2: While authors showed that E2-KLH conjugate had highest anti-prorenin antibody titer, this peptide is 11 amino acid long and could potentially activate autoreactive T-cells. T-cell response should be evaluated. IgG subclass of peptide-induced antibodies and levels of complement system proteins should also be investigated.

Response 2: We very much appreciate your comment, especially from the view of safety concerns. As you point out, we evaluated whether the antigen peptide (E2) itself activates T-cells in vaccinated mice, as assessed by the ELISPOT assay. As shown in the new Figure 8, KLH treatment induced IFN- *γ* expression, but prorenin recombinant treatment did not, suggesting that E2 peptide does not include the T-cell epitope and therefore does not activate T cells. We also evaluated the IgG subclass of vaccine-induced antibody, and the results suggested that Th2 (mouse IgG1) is dominant (Supplementary Figure 2). Because mouse IgG1 does not have Fc-dependent action, we speculate that the induced antibody does not activate the complement pathway. We have added new text at lines 205-217 in the Methods, lines 285-293 in the Results, and lines 386-393 in the Discussion.

Comment 3: The experimental design for in vivo efficacy study of the peptide vaccine lacked the db/m + Vp treated control group.

Response 3: Thank you so much for your suggestion. Actually, we performed ERG also in db/m + VP. The reason why we excluded the ERG data of db/m + VP in the first submission was that we have not yet performed the flicker and high O2 tests. When we performed these tests in our previous work (Hanaguri et al, Sci Rep 2021), we found it almost impossible to perform all these procedures in this series. Therefore, we decided to keep the tests that we performed in the three groups to a minimum to efficiently prove the efficacy of prorenin peptide vaccine. In the current version of the manuscript, we have added only the ERG data of the implicit times of the b-wave and ΣOPs in db/m + VP compared with db/m + KLH, as shown in Supplementary Figure 1. The experimental design for db/m + VP is provided as Supplementary Material 1, and the results are presented in lines 274-277.

Comment 4: In all figures with immunostaining, there are no quantitative measurements and statistical analysis (Fig 3H, Fig 4C, Fig 5F, Fig 6).

Response 4: Thank you so much for your suggestion. We now show the quantitative measurements and results of the statistical analysis of the intensity of immunofluorescence for each staining except for DAB staining for p-ERK; we are unable to show the analysis for p-ERK staining because we could not detect sufficient fluorescein staining to enable us to quantitate the amount of p-ERK in the retina. If we want to quantitate p-ERK in the retina in a future study, we will need to prepare the whole retina sample for Western blotting. 

Comment 5: As elevated prorenin in the retina is thought to contribute pathogenesis of diabetic retinopathy, the levels of prorenin/renin and Ang I/II in the retina with and without treatments should be measured. This is important as authors showed that this peptide vaccine seemed to only block prorenin-induced pathway(s) independent of Ang II, without affecting circulating renin level and it is important to know whether the beneficial effects of this peptide vaccine is mediated via systemic effects or locally via blocking prorenin and its receptor activation. It would be also interesting to compare the efficacy of the prorenin peptide vaccine with classic RAS blockers such as ACE inhibitors and Ang II receptor blockers.

Response 5: We completely agree with your suggestion. As mentioned in the limitations, we did not show the retinal levels of prorenin in diabetic db/db mice. In fact, in the current study we attempted to perform an immunohistochemical analysis for prorenin and (pro)renin receptor, but we obtained results only for immunohistochemistry of the (pro)renin receptor. An alternative way to detect prorenin in the retina is PCR. Therefore, we have added a sentence to the limitations that mentions the need for a future study that uses PCR to directly demonstrate the expression of prorenin in the retina of this type 2 diabetic model (lines 409-411). 

 To study the effect of prorenin peptide vaccine by angiotensin II blockade, we used angiotensin II peptide vaccine (VA) instead of either conventional angiotensin II receptor blockers or angiotensin II converting enzyme inhibitors. We used this approach because we were focusing on the benefits of peptide vaccine therapy, such as low cost and, unlike the situation with daily medication, lack of concerns about patient adherence. We now present the ERG and antibody titer data in the new Figure 9. Surprisingly, angiotensin II peptide vaccine showed absolutely no protective effect even though the antibody titer was almost the same as that we found in our previous work, which showed an anti-hypertensive effect and organ protection in various models. Therefore, at least from these results, we can conclude that in terms of peptide vaccine for diabetic retinopathy, prorenin is a more suitable target molecule than angiotensin II. We have added the VA results at lines 294-300 in the Results and lines 394-402 in the Discussion. As mentioned in the limitations, we did not show the local expression of prorenin and angiotensin II in the retina of diabetic db/db mice. 

Comment 6: In the Introduction and Discussion, there is insufficient discussion on related background literature, why prorenin is an ideal target molecule in the prevention of diabetic retinopathy, some important early reports on prorenin in diabetic retinopathy are not mentioned. There is insufficient discussion on data presented and potential mechanisms.

Response 6: Thank you for your suggestion. To make it easier for readers to understand why the ideal target molecule is prorenin and not angiotensin II, we have added sentences at lines 60-64 in the Introduction. In the Discussion, we have cited previous works that demonstrate a significant role of prorenin in microglial activation (lines 415-422).

Please note that during the revision process, we needed to make further minor changes; these changes are highlighted in yellow. 

Responses to Reviewer 2

Comment 1: This is an interesting manuscript because it demonstrates that vaccination may be an alternative approach to prevent diabetic retinopathy. Furthermore, the authors report that T2D severely affects the retinal blood flow response to hyperoxia and flicker light stimulation. Moreover, the efficacy of the prorenin-directed vaccine indicates that prorenin contributes to glial dysfunction and early pathogenesis of diabetic retinopathy. Finally, they speculate that microglial activation may compromise neurovascular coupling and thereby promote retinopathy.

Response 1: Thank you very much for your positive remarks. We have carefully followed your suggestions, as described below. 

Comment 2: While the text states that there is no difference in the prorenin receptor level, Fig 3H shows that there is less in the retinal from the vaccinated animal. Please quantify the results shown in Fig 3H.

Response 2: Thank you so much for this suggestion. We have quantified the fluorescence intensity and changed the images as well.

Comment 3: Please modify Fig 4A and B to indicate the times at which there are statistically significant differences between the experimental groups.

Response 3: Thank you for this suggestion. The difference between db/m + KLH and db/db + KLH was significant, and significance is now indicated in the figure with asterisks. In addition, the difference between db/db + KLH and db/db + VP also was significant, and this significance is indicated with daggers. 

Comment 4: Please quantify the results shown in Fig 4C.

Comment 5: Please quantify the results shown in Fig 5F.

Comment 6: Please quantify the results shown in Fig 6.

Response 4, 5, 6: Thank you so much for these suggestions. We have quantified the intensity of immunofluorescence in all these figures.

---

## [Decision Letter · Decision Letter 1]

30 Dec 2021

Effect of prorenin peptide vaccine on the early phase of diabetic retinopathy in a murine model of type 2 diabetes

PONE-D-21-16317R1

Dear Dr. Yokota,

We’re pleased to inform you that your manuscript has been judged scientifically suitable for publication and will be formally accepted for publication once it meets all outstanding technical requirements.

Kind regards,

Alfred S Lewin, Ph.D.

Section Editor

PLOS ONE

Additional Editor Comments (optional):

Reviewers' comments:

Reviewer's Responses to Questions

**Comments to the Author**

1. If the authors have adequately addressed your comments raised in a previous round of review and you feel that this manuscript is now acceptable for publication, you may indicate that here to bypass the “Comments to the Author” section, enter your conflict of interest statement in the “Confidential to Editor” section, and submit your "Accept" recommendation.

Reviewer #2: All comments have been addressed

2. Is the manuscript technically sound, and do the data support the conclusions?

Reviewer #2: (No Response)

3. Has the statistical analysis been performed appropriately and rigorously? 

Reviewer #2: (No Response)

4. Have the authors made all data underlying the findings in their manuscript fully available?

Reviewer #2: (No Response)

5. Is the manuscript presented in an intelligible fashion and written in standard English?

Reviewer #2: (No Response)

6. Review Comments to the Author

Reviewer #2: (No Response)

7. PLOS authors have the option to publish the peer review history of their article (what does this mean?). If published, this will include your full peer review and any attached files.

Reviewer #2: No

---

## [Editor Report · Acceptance letter]

7 Jan 2022

PONE-D-21-16317R1 

Effect of prorenin peptide vaccine on the early phase of diabetic retinopathy in a murine model of type 2 diabetes 

Dear Dr. Yokota:

I'm pleased to inform you that your manuscript has been deemed suitable for publication in PLOS ONE. Congratulations! Your manuscript is now with our production department. 

Kind regards, 

on behalf of

Dr. Alfred S Lewin 

Section Editor

PLOS ONE